# Light-Induced Flavonoid Biosynthesis in *Sinopodophyllum hexandrum* with High-Altitude Adaptation

**DOI:** 10.3390/plants12030575

**Published:** 2023-01-28

**Authors:** Qiaozhu Zhao, Miaoyin Dong, Mengfei Li, Ling Jin, Paul W. Paré

**Affiliations:** 1College of Life Science and Technology, Gansu Agricultural University, Lanzhou 730070, China; 2State Key Laboratory of Arid Land Crop Science, Gansu Agricultural University, Lanzhou 730070, China; 3College of Pharmacy, Gansu University of Chinese Medicine, Lanzhou 730101, China; 4Department of Chemistry and Biochemistry, Texas Tech University, Lubbock, TX 79409, USA

**Keywords:** *Sinopodophyllum hexandrum*, high altitude, flavonoid biosynthesis, gene expression, high light

## Abstract

*Sinopodophyllum hexandrum* is a perennial alpine herb producing the anti-cancer metabolite podophyllotoxin (PPT). Although the adaptation of *S. hexandrum* to high altitudes has been demonstrated and the effects of temperature, precipitation, and UV-B light on plant growth and metabolite accumulation have been studied, knowledge on the role of flavonoid biosynthesis in adapting to high altitudes is limited. In this study, light intensity, amount and type of flavonoids, and differentially expressed proteins (DEPs) and genes (DEGs) at 2300 and 3300 m were analyzed by HPLC, proteomic, transcriptomic, and qRT-PCR analysis. We found that higher light intensity correlated with greater flavonoid, flavonol, and anthocyanin content as well as higher anthocyanin to total flavonoid and flavonol ratios observed at the higher altitude. Based on proteomic and transcriptomic analyses, nine DEPs and 41 DEGs were identified to be involved in flavonoid biosynthesis and light response at 3300 m. The relative expression of nine genes (*PAL*, *CHS1*, *IFRL*, *ANS*, *MYB4*, *BHLH137*, *CYP6*, *PPO1*, and *ABCB19*) involved in flavonoid biosynthesis and seven genes (*HSP18.1*, *HSP70*, *UBC4*, *ERF5*, *ERF9*, *APX3*, and *EX2*) involved in light stress were observed to be up-regulated at 3300 m compared with 2300 m. These findings indicate that light intensity may play a regulatory role in enhancing flavonoid accumulation that allows *S. hexandrum* to adapt to elevated-altitude coupled with high light intensity.

## 1. Introduction

*Sinopodophyllum hexandrum* Royle (family Berberidaceae) is a perennial rhizomatous species that is native to the alpine Himalayan region at altitudes of 2000 to 4500 m above sea level [1,2,3]. The dried fruit is referred to as “*xiaoyelian*” in China and is used widely as a traditional Tibetan medicine to treat gynecological diseases [4]. The rhizomes are the major source of PPT, which can effectively mitigate specific cancers and heal certain skin lesions [5]. Constituents from dried fruit and rhizomes include lignans, flavonoids, and alkaloids [4,5,6]. The species is currently endangered due to over exploitation of wild plants and limited large-scale artificial cultivation [7,8].

To protect the wild *S. hexandrum* and to satisfy the commercial demand, preliminary preparations for large-scale cultivation have been performed, including breaking seed dormancy [9], promoting seed germination [10,11], accelerating vegetative propagation by dividing the rhizomes [9], and providing the sustainable cultivation–collection model for optimized PPT production [3]. Currently, large-scale cultivation has yet to be realized, largely because of a failure to identify suitable growth conditions in the field.

Geographical modeling for *S. hexandrum* predicts that altitude determines plant distribution, with optimal altitudes of 2800 to 3600 m for artificial planting [2]. There is a significant positive correlation of increasing altitude with plant biomass and PPT accumulation [1,3,12,13,14,15]. However, other environmental factors such as temperature, precipitation, and light may also affect plant growth [1,2,3,14]. To dissect the effect of individual environmental components on growth, studies are best executed under controlled laboratory conditions. What is reported thus far is that cool temperatures and reduced water availability are optimal for plant growth [16,17,18,19], while elevated UV-B radiation and high light intensity inhibit growth [20,21].

Flavonoids (i.e., flavonols, flavones, isoflavones, anthocyanidins, flavanones, flavanols, and chalcones) play an essential role in response to elevated light [22,23,24]. Specifically, full sunlight promotes quercetin, kaempferol, and isorhamnetin biosynthesis in ginkgo leaves [25], while petunia leaves and stems under elevated light promote anthocyanin accumulation [26]. Visible light induces proanthocyanidin accumulation in grapes, while UV radiation induces flavonol accumulation in young berries [27]. In fact, anthocyanins can protect plants against light stress by maintaining the higher oxidative levels and minimizing damage to photosynthetic tissues [28].

At the cellular level, flavonoids are synthesized in the cytoplast and then transported into the vacuole to be stored [29]. Generally, flavonoid biosynthesis is regulated by transcription factors (TFs) including MYB, bZIP, bHLH, or their complex, as well as by cytochrome P450s (CYPs) [22,30,31]. Flavonoids are transported through two different transport mechanisms: proton antiport and ABC-type transporter in various plant species [32]. Finally, the stable flavonoids are stored after the modification of acylation (e.g., acetyltransferase, ATs), methylation (e.g., methyltransferases, MTs), and glycosylation (e.g., UDP-glycosyltransferases, UGTs) [29,33].

Previous studies on the developmental adaptation of *S. hexandrum* have found that anthocyanins in dark spots on the leaf can function as UV protectants, effectively absorbing UV-B radiation at high altitudes [34,35]. In our previous findings, the total flavonoid content was observed in *S. hexandrum* seedlings under UV-B treatment [21]; a total of 234 characterized genes were differentially expressed at 3300 versus 2300 m, with 22 genes involved in flavonoid biosynthesis and 19 genes involved in light response [36]. To date, the role of flavonoid biosynthesis in adapting to high altitudes is limited. In this study, light intensity and total flavonoids and DEPs were analyzed by spectrophotometer, HPLC, and proteomic approaches. The DEGs were re-analyzed based on our previous transcriptomic data and the expression levels of genes involved in flavonoid biosynthesis and light stress were validated by qRT-PCR. A significant difference in flavonoid accumulation as well as protein and gene expression associated with flavonoid biosynthesis was observed with elevated light.

## 2. Results

### 2.1. Differences in Light Intensity and Flavonoid Accumulation

Total flavonoid content was slightly elevated (1.08-fold greater, Figure 1A) for plants grown at 3300 m compared with 2300 m. The light intensity was 1.25-fold greater at the higher altitude (Figure 1B). Specifically, rutin, quercetin, and kaempferol contents were 1.28-, 2.83-, and 2.11-fold greater, while the isoquercetin exhibited a 0.34-fold decrease (Figure 1C). The total content of the four flavonols (rutin, quercetin, kaempferol, and isoquercetin) showed a 1.04-fold increase (Figure 1D) and the anthocyanin content was 2.09-fold greater (Figure 1E).

### 2.2. Ratio Differences for Anthocyanins and Total Flavonoids

The ratio of anthocyanins to total flavonoid content was 1.91-fold higher (Figure 2A) and the ratio of anthocyanins to four flavonols’ contents was 1.99-fold higher at 3300 m than 2300 m (Figure 2B).

### 2.3. Differentially Expressed Proteins at Higher Elevation

A total of 65 proteins were differentially expressed at 3300 m vs. 2300 m, with 46 over-expressed and 19 down-expressed (Appendix A); among the 46 over-expressed proteins, 30 proteins had known function, while the other 16 proteins had no known function (Table 1). Proteins were classified based on biological function, including primary and secondary metabolism (6), photosynthesis and energy (10), cell morphogenesis (2), transcription (4), and translation (8). Among 30 proteins grouped by biological function, 5 proteins were involved in flavonoid biosynthesis (ANS, CYP6, CYP71BE30, PPO1, and MYB4) and 4 proteins were involved in photosynthesis (ycf3, rbc, rbcL, and petL).

### 2.4. Differentially Expressed Genes Based on Elevation Differences

Transcriptomic data of plants at 2300 and 3300 m [36] showed that 22 genes encoding for flavonoid catalysis were differentially expressed, with 16 genes up-regulated 1.4-(*C4H*, *ABCB19,* and *CYP71CU1*) to 4.9-fold (*PPO1*), while 6 other genes were down-regulated by 1.6- (*DTX41*) to 5.0-fold (*CHS1*) (Table 2). In addition, 19 genes involved in light response were differentially expressed, with 9 genes associated with photosynthesis and 10 genes mediated by light stress (Table 3).

### 2.5. Expression Level of Genes Involved in Flavonoid Biosynthesis

Of the five DEPs (Table 1) and 22 DEGs (Table 2) involved in flavonoid biosynthesis, eight genes (*PAL*, *CHS1*, *GT6*, *IFRL*, *BHLH137*, *ABCB19*, *CYP6*, and *PPO1*) and four proteins (ANS, MYB4, CYP6, and PPO1) were identified with CYP6 and PPO1, exhibiting overlap for both gene and protein differential expression at the higher altitude. These subsets of differentially expressed genes/proteins were selected to measure relative expression levels (RELs) by qRT-PCR. A 3.3- (*ABCB19*) to 12.3-fold (*PAL*) up-regulation and 0.5-fold (*GT6*) down-regulation were observed at the higher altitude (Figure 3). These RELs were consistent with the fold change (FC) at the two altitudes, except for the *CHS1* (Table 2).

### 2.6. Expression Level of Genes Involved in Light Stress

To further examine the biological function of genes involved in light stress [36], seven genes (*HSP18.1*, *HSP70*, *UBC4*, *ERF5*, *ERF9*, *APX3*, and *EX2*) were monitored for the REL by qRT-PCR (Table 3). A 2.03- (*UBC4*) to 15.47-fold (*ERF5*) up-regulation was observed at 3300 vs. 2300 m (Figure 4), which was consistent with the FC at the two altitudes, except for the *UBC4* (Table 3).

## 3. Discussion

The alpine medicinal plant *S. hexandrum* has been demonstrated in previous studies to be adapted to high-altitude conditions [1,2,3]. Although anthocyanin dark spots on the leaf surface were proposed to function as UV protection at high altitudes [34,35], the induction of flavonoid biosynthesis by high altitude conditions had not been investigated. Here, we report that higher light intensity is associated with greater flavonoid, flavonol, and anthocyanin content and that higher ratios of anthocyanins to total flavonoids and flavonols are observed for plants grown at 3300 m. Indeed, nine proteins and 41 genes directly involved in flavonoid biosynthesis and light response are differentially expressed in plants grown at the higher altitude.

Previous studies found that climatic factors (e.g., temperature, precipitation, and light) contributed more to elevated lignan and flavonoid content in *S. hexandrum* than soil elements (e.g., pH, organic matter, and potassium). Specifically, low temperatures (4–15 °C) could enhance plant biomass and accumulation of secondary metabolites, especially in podophyllotoxin (PPT), in *S. hexandrum* plants by the up-regulation of genes involved in plant growth, PPT biosynthesis, and stress response [16,17,18]; a moderate water deficit could improve PPT accumulation [19]; a moderately shaded habitat could enhance plant growth [20]; UV-B radiation could inhibit plant growth and PPT biosynthesis while improving flavonoid accumulation [21]. Among other effects, the lower temperatures expected in the higher altitude site could probably negatively affect the dark phase of the photosynthesis, therefore exacerbating the photo damage and ROS production caused by high light intensity.

Previous studies have found that higher altitudes (Shangri-La of Yunnan Province and Nyingchi of Tibet) seem to be essential for the accumulation of flavonoids (e.g., quercetin and kaempferol) [14]. Initially, in situ climatic factors including lower annual air temperature, lower annual light duration, and higher annual precipitation at higher elevation growing sites were considered as drivers for robust *S. hexandrum* growth [3]. However, an alternative driver for growth is significantly higher light intensity at the higher elevation. Indeed, correlation analysis showed a positive correlation between annual light duration and kaempferol and quercetin content [14]. In this study, greater total flavonoids, flavonols, and anthocyanins were observed and correlated with higher light intensity measured at 3300 m than the lower elevation. Light quality and intensity have previously been shown to be a driver of flavonoid biosynthesis in plants [23,24,25,26,27]. In fact, the UV-B radiation enhancing total flavonoid accumulation in *S. hexandrum* seedlings has been observed [21].

Other climatic factors such as temperature, precipitation, and air pressure may also contribute to flavonoid biosynthesis and/or accumulation at high altitudes [1,2,3]. Several genes involved in upstream (*PAL*, *C4H*, and *4CL*) and downstream (MTs and UGTs) flavonoid biosynthesis, as well as TFs (MYB, bHLH, and WRKY) and CYPs under low-temperature and water deficit treatments [16,19], have been published; climatic measurements to directly link temperature and/or precipitation with enhanced flavonoid accumulation have yet to be reported.

In this study, five proteins (ANS, CYP6, CYP71BE30, PPO1, and MYB4) and 22 genes (*PAL*, *C4H*, *CHS1*, *IFRL*, *GT6*, *BHLH137*, *DTX41*, *ABCB19*, nine *CYPs*, *PPO1*, and four *MTs*) may be involved in flavonoid biosynthesis. Extensive studies have demonstrated that PAL, C4H, CHS1, ANS, and GT6 directly participate in flavonoid biosynthesis, the ABCB19 participates in flavonoid transport, and the MYB4 and BHLH13 participate in the regulation of flavonoid biosynthesis [29,30,37,38,39,40,41,42]. In addition, nine CYPs and four MTs may also participate in flavonoid biosynthesis [31,33,43]. In addition, the PPO1 and IFRL are involved in the biosynthesis of anthocyanins and isoflavanones, respectively [44,45], and the DTX41 acts as a flavonoid/H^+^-antiporter controlling the flavonoid transport [46]. It is noteworthy that two TFs, MYB4 and BHLH137, may play critical roles in flavonoid biosynthesis. As is known, the R2R3-MYBs participate in plant growth and development, metabolism (e.g., flavonoid biosynthesis), and stress responses [47]. Previous studies have found that MYB4 serves as a repressor for flavonoid biosynthesis and that *MYB4* expression is down-regulated and can enhance sinapate ester levels in *Arabidospis* leaves with exposure to UV-B light [48]. In buckwheat, in response to UV-B, *MYB4R1* regulates flavonoid and anthocyanin biosynthesis by binding to L box motifs in the promoter of *CHS*, *FLS*, and *UFGT* [49]. Moreover, the MYB-bHLH-WD40 complex can regulate genes that encode late step enzymes in the pathway, leading to increased flavonoid biosynthesis [50]. Based on the biological function, the roles of identified proteins and genes were mapped in the flavonoid biosynthetic pathway (Figure 5).

The difference in *CHS1* gene expression levels based on transcriptomic and qRT-PCR analysis may result from the annotation of the reference transcriptome, amplification efficiency, and/or individual features. Previous studies have found that there are about 85% genes of RNA sequencing consistent with qRT-PCR data [51]. For the down-regulation of the *GT6* gene, previous studies on strawberry found that GT6 might be associated with the glucosylation of flavonols [52]; thus, its down-regulation may inhibit the glucosylation of flavonols, which is in accordance with the results of greater contents of flavonols (rutin, quercetin, and kaempferol) at 3300 m than 2300 m.

Initially, flavonoid biosynthesis is regulated by biotic and abiotic stresses that include light stress [53,54]. Here, 10 DEGs (*HSP18.1*, *HSP90-1*, *HSP22.0*, *HSP70*, *UBC4, ERF5*, *ERF9*, *TLP*, *APX3*, and *EX2*) were identified to be involved in light stress. Generally, the HSPs function to prevent damage under heat, high light, and free radicals [55,56]; meanwhile, the up-regulation of *HSPs* can increase flavonoid content [57]. UBC4 can enhance the flavonoid accumulation by regulating the expression of the *CHS* gene [58,59]. For other genes, ERFs are involved in the regulation of gene expression by stress factors [60]; TLP can improve plant tolerance [61]; APX3 may be involved in the detoxification of H_2_O_2_ [62]; EX2 together with EX1 enables plants to perceive stress signal [63]. In this study, the up-regulation of selected genes (*HSP18.1*, *HSP70*, *UBC4, ERF5*, *ERF9*, *APX3*, and *EX2*) may play important roles in enhancing flavonoid biosynthesis and protecting plants from abiotic stress (e.g., high light) at high altitudes. In addition, elevated expression *CAB8*, *CAB21*, and *PSBR,* as well as proteins ycf3, rbc, rbcL, and petL, at 3300 m versus 2300 m (Table 1 and Table 3) suggest that flavonoid accumulation can protect PSII and PSI from stronger UV radiation, which could explain greater biomass that is observed for plants grown at the higher elevation [3].

Based on the above results, a model of high-altitude-induced flavonoid biosynthesis in *S. hexdanrum* is proposed (Figure 6). When plants are exposed to high altitudes and high light intensity, genes involved in light stress (e.g., *HSPs*, *ERFs*, and *TLP*) are up-regulated. Photosynthesis will initially be inhibited, as observed with the down-regulation of genes (e.g., *CABs*, *RBCs*, and *TPT*), and subsequently, flavonoids (e.g., flavonols and anthocyanins) will be synthesized and accumulated with the up-regulation of genes (e.g., *PAL*, *CHS1*, and *CYPs*). Additionally, the ratio of anthocyanins will be increased with the up-regulation of genes (e.g., *ANS* and *MYB*) and, finally, the anthocyanin pigments in leaves will play critical roles in absorbing the high light radiation (e.g., UV-B) and removing the free radicals (e.g., H_2_O_2_), which confers the ability of the plants to adapt to the high altitude environmental conditions. 

## 4. Materials and Methods

### 4.1. Plant Materials

Mature fruit of *Sinopodophyllum hexandrum* were harvested in September 2012 from Gannan Tibetan Autonomous Prefecture (3300 m a.s.l.; 34°42′53″ N, 102°53′12″ E) of Gansu province, China. The seeds separated from the fruit were pre-treated with 0.01% Bavistin for 10 min, immersed in sterile 500 mg/L GA_3_ for 48 h, and then sown at the 3300 m sites in May 2013. Some seedlings were transplanted to Weiyuan (2300 m; 34°58′8″ N, 104°04′52″ E) of Gansu province in May 2014 [3]. The environmental parameters and soil characteristics were shown in Appendix A, respectively.

*S. hexandrum* leaves from 3-year-old plants were collected at 12:00 to 14:00 pm on a sunny day from the 2300 and 3300 m sites in July 2016, with 30 leaves at each altitude pooled (×3) and divided into two parts after mixing; one part was rapidly frozen in liquid nitrogen for proteomic partly and the other part was air dried at room temperature for flavonoid analysis (Figure 7). The light intensities at 2300 and 3300 m were measured at 13:00 pm during sunny days (in early July 2016) using the agricultural meteorology monitor (TNHY-7; Zhejiang Tuopu Instrument Co., Ltd., Hangzhou, China).

### 4.2. Determination of Total Flavonoid, Flavonol, and Anthocyanin Contents

#### 4.2.1. Extract Preparation

Air-dried leaf powder (0.3 g) was soaked in ethanol (20.0 mL, 95% *v*/*v*) at 22 °C for 72 h and then centrifuged at 6000 rpm at 4 °C for 10 min. Following exhaustive extraction (×3), the supernatant was increased to 20 mL with 95% ethanol and concentrated in a rotary evaporator at 55 °C. The concentrated residue was re-dissolved with 9 mL methanol [17,18].

#### 4.2.2. Determination of Total Flavonoid Content

Extracts (150 µL) were added into ddH_2_O (2 mL) and 5% NaNO_2_ (0.3 mL). After agitating for 5 min, 10% AlCl_3_ (0.3 mL) was added and reacted for 1 min and then 1.0 mol/L NaOH (2 mL) was added to stop the reaction. Absorbance was taken at 510 nm [18]. The total flavonoid content was calculated based on the standard curve (Appendix A).

#### 4.2.3. Flavonol Quantification

Extracts filtered with a 0.22 μm Durapore membrane were analyzed at 365 nm using an HPLC (Eclipse Plus C18, 250 mm × 4.6 mm, 5 μm, column temperature 30 °C). Acetonitrile (A): phosphoric acid (0.1%*v*/*v*, B) was the mobile phase with gradient elution: 15–40% A (0–8 min), 40–65% A (8–12 min), 65–85% A (12–14 min), 85–15% A (14–16 min), and 15% A (16–20 min) at a flow rate of 1.0 mL/min [64]. The contents of rutin, isoquercetin, quercetin, and kaempferol were calculated based on peak area comparison with reference standards (Appendix A).

#### 4.2.4. Determination of Anthocyanin Content

Air-dried leaf powder (0.5 g) was soaked in methanol (5.0 mL, 0.1% HCL *v*/*v*) at 22 °C for 72 h, then centrifuged at 5000 rpm at 4 °C for 30 s. Following exhaustive extraction (×3), the supernatant was increased to 20 mL with methanol (0.1% HCL *v*/*v*). Absorbance was taken at 530 nm and the anthocyanin content was calculated based on a relative level compared with the blank control [33,65].

### 4.3. Transcriptomic Analysis

Transcriptomic analysis was performed as previously described [36]. For the methodology of RNA-seq, total RNA samples were isolated using Trizol reagent (Invitrogen, Carlsbad, CA, USA) and purified using RNase-free DNase I (TakaRa, Dalian, China); poly-A mRNA was enriched and then used to prepare the paired-end cDNA library (2 × 126 nt) (Illumina, San Diego, CA, USA). Sequences were trimmed with 5′ and 3′ prime ends to remove poor-quality reads; paired-end sequencing was performed using an Illumina Hiseq 2000 platform. De novo assembly was carried out using Trinity software (version 2.0.6) [66]. Unigenes were searched against a NR, Swiss-Prot, TrEMBL, Pfam, and KOG using a BLASTx procedure with an e-value ≤ 10^−5^ [36,67]; DEGs were identified with |log_2_(fold-change) ≥ 1 and *p* ≤ 0.05 by DESeq2 software and the edgeR package [68,69].

### 4.4. Proteomic Analysis

#### 4.4.1. Protein Extraction

Proteins were extracted according to a previously described method [70]. Firstly, frozen leaves were pulverized to a fine powder in liquid nitrogen. Aliquots (200 μL) were then mixed with TCA-2ME-acetone solution (1.8 mL of 10% TCA (*w*/*v*), 0.07% 2ME (*v*/*v*) in cold acetone) and stored at −20 °C for 1 h. The homogenate was then centrifuged at 10,000× *g* and 4 °C for 10 min and the precipitate was re-suspended in cold acetone (1.8 mL containing 0.07% 2ME (*v*/*v*)). The mixture was then stored at −20 °C for 1 h and then centrifuged at 10,000× *g* and 4 °C for 15 min. This step was repeated at least three times until the supernatant became colorless. The precipitate was vacuum dried, dissolved in a protein solubilization buffer (1.8 mL of 5 M urea, 2 M thiourea, 2% CHAPS (*w*/*v*) with 2% *N*-decyl-*N*,*N*-dimethyl-3-ammonio-1-propane sulfate (*w*/*v*), 20 mM DTT, 5 mM phosphine, 0.5% pharmalyte pH 4–6.5 (*v*/*v*), and 0.25% pharmalyte pH 3–10 (*v*/*v*) in ddH_2_O) and then vortexed for 1 min. Finally, the solubilized mixture was centrifuged at 10,000× *g* and 25 °C for 15 min and then the supernatant was centrifuged to remove any cellular debris. Protein amount was quantified using a 2-D Quant kit (GE Healthcare, Wauwatosa, WI, USA) and the protein extracts were stored at −80 °C.

#### 4.4.2. 2-DE Separation

Firstly, protein samples were centrifuged at 10,000× *g* and 25 °C for 15 min and then diluted in rehydration buffer (8 M urea, 2% CHAPS (*w*/*v*), 0.3% DTT (*w*/*v*), and 1% IPG buffer (*v*/*v*)). Secondly, IEF strips (24 cm, linear pH 4–7) were re-hydrated in protein solution (450 μL at 20 °C) for 12 h in an electrophoresis chamber (Ettan IPGphor 3 system; GE Healthcare, Wauwatosa, WI, USA). Thirdly, the IEF was performed using the following parameters: 200 V for 2 h, 500 V for 2 h, 1000 V for 2 h, and 8000 V for 4 h. Fourthly, after the IEF, strips were incubated twice in equilibration buffer (8 M urea, 1.5 M Tris-HCl buffer (pH 8.8), 30% glycerol (*v*/*v*), and 2% SDS (*w*/*v*)) with 1% DTT (*w*/*v*) for 15 min, followed by 4% (*w*/*v*) of iodoacetamide for another 15 min. Fifthly, an Ettan DALTSix system (GE Healthcare, Wauwatosa, WI, USA) was used for 2-DE with the strips placed on a 12.5% SDS-PAGE gel (*w*/*v*) and run at 1 W for 1 h and then at 12 W for 5 h per gel. Subsequently, gels were fixed in 40% alcohol (*v*/*v*) and 10% acetic acid (*v*/*v*) for 30 min, swollen in 10% acetic acid (*v*/*v*) for 20 min, and stained with CBB G-250. Finally, the stained gels were rinsed with 10% acetic acid (*v*/*v*) until protein spots were distinct from the background and stored in deionized water (Appendix A).

#### 4.4.3. Gel Scanning and Image Analysis

The stained gels were scanned using an ImageScanner III (GE Healthcare, Wauwatosa, WI, USA) and images were analyzed using ImageMaster 2-D Platinum v7.0 software (GE Healthcare, Wauwatosa, WI, USA), with three independent biological replicates for the 2300 or 3300 m samples. Spots were automatically detected and matched; mismatched and unmatched spots were artificially modified through manual editing. Spot densities were expressed as mean normalized volumes, and fold changes were calculated between 2300 and 3300 m. Only spots with intensity ratios over 1.5-fold were selected as differentially expressed proteins (DEPs) and the over-expressed proteins at 3300 m vs. 2300 m were subsequently identified.

#### 4.4.4. Protein Identification

The selected DEP spots were excised from CBB-stained gels and digested with trypsin. Peptides were identified using an AB SCIEX 5800 MALDI-TOF/TOF^TM^ system (AB SCIEX, CA, USA) according to a previously described protocol [71]. The obtained peptide masses were used to search the NCBI database using a MASCOT engine (http://www.matrixscience.com (accessed on 1 June 2020)). Biological functions of identified proteins were classified using the Swiss-Prot database (http://www.uniprot.org (accessed on 6 November 2021)). After removing the repeat proteins by selecting the longer sequencing of amino acids for the same protein, the DEPs involved in light response and flavonoid biosynthesis were further screened and validated in this study.

### 4.5. qRT-PCR Validation of Genes Involved in Light Stress and Flavonoids Biosynthesis

Based on the biological functions of the DEGs and DEPs, 17 representative genes were selected to validate their expression level by qRT-PCR. The validation of genes was performed according to our previous protocol [21]. The cycle threshold (*Ct*) values and standard curves of the reference gene *actin* (*ACT*) at different concentrations (0.25, 0.5, 1.0, 1.5, 2.0, and 3.0 μL) were performed to correct expression level of selected genes (Appendix A). The primer sequence (Appendix A) was designed using the primer-blast tool in NCBI and synthesized by Sangon Biotech (Shanghai, China). Total RNA samples at 2300 and 3300 m were extracted from the leaves using a Plant RNA Kit (R6827; Omega Bio-Tek, Norcross, GA, USA). cDNA was synthesized using a FastKing RT Kit (KR116; Tiangen, Beijing, China) with 42 °C for 15 min and 95 °C for 3 min for one cycle. The qRT-PCR was carried out using a SuperReal PreMix (FP205; Tiangen, Beijing, China) at 95 °C (15 min) for one cycle, followed by 95 °C (10 s), 60 °C (20 s), and 72 °C (30 s) for 40 cycles. The melting curve was analyzed at 72 °C for 34 s. TheREL was calculated using a 2^−∆∆*Ct*^ method [72].
∆*Ct*
_Test gene_ = *Ct* _Test gene_ − *Ct* _Reference gene_
∆*Ct*
_Control gene_ = *Ct* _Control gene_ − *Ct*
_Reference gene_
−∆∆*Ct*
_(3300 vs. 2300 m)_ = −(∆*Ct*
_3300 m_ − ∆*Ct*
_2300 m_)
REL = 2^−∆∆*Ct*^

### 4.6. Statistical Analysis

All measurements were performed using three biological replicates. Statistical analysis was conducted via a *t*-test for independent samples. SPSS 22.0 was the software package used, with *p* < 0.05 as the basis for statistical differences.

## 5. Conclusions

From the above observations, greater contents of total flavonoids, flavonols, and anthocyanins in *S. hexandrum* were observed at higher altitudes, which may be regulated by the up-regulated genes or over-expressed proteins involved in flavonoid biosynthesis and light stress. These findings indicate that light stress may play a determining role in enhancing flavonoid accumulation at high altitudes. In order to exclude the differences in temperature, water, and other factors, the role of light in affecting plant growth and flavonoid biosynthesis will be further studied under controlled laboratory conditions such as shaded habitats.

## Figures and Tables

**Figure 1 plants-12-00575-f001:**
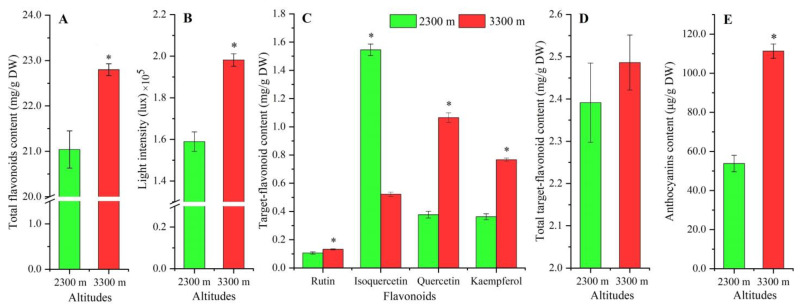
The content of total flavonoids (**A**), light intensity (**B**), content of target flavonoid (**C**), total target flavonoid (**D**), and anthocyanins (**E**) at 2300 and 3300 m. The * represents a significant difference (*p* < 0.05) between 2300 and 3300 m (mean ± SD, *n* = 3).

**Figure 2 plants-12-00575-f002:**
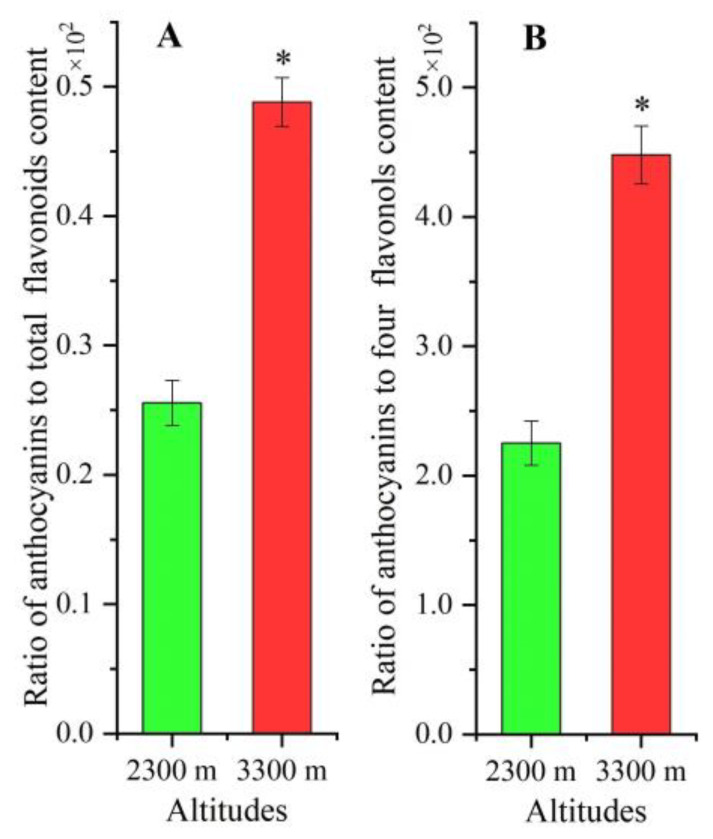
The ratios of anthocyanins to total flavonoids (**A**) and flavonol contents (**B**) at 2300 and 3300 m. The * represents a significant difference (*p* < 0.05) between 2300 and 3300 m (mean ± SD, *n* = 3).

**Figure 3 plants-12-00575-f003:**
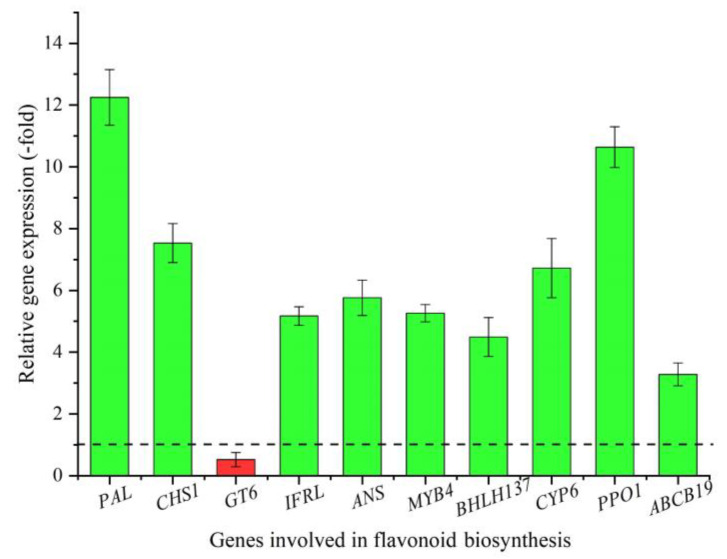
The REL of genes involved in flavonoid biosynthesis at 3300 vs. 2300 m, as determined by qRT-PCR (mean ± SD, *n* = 3). The dotted line in the image differentiates up-regulation (>1) and down-regulation (<1) at 3300 vs. 2300 m, respectively. The columns highlighted in green represent up-regulation and red represents down-regulation. The same below.

**Figure 4 plants-12-00575-f004:**
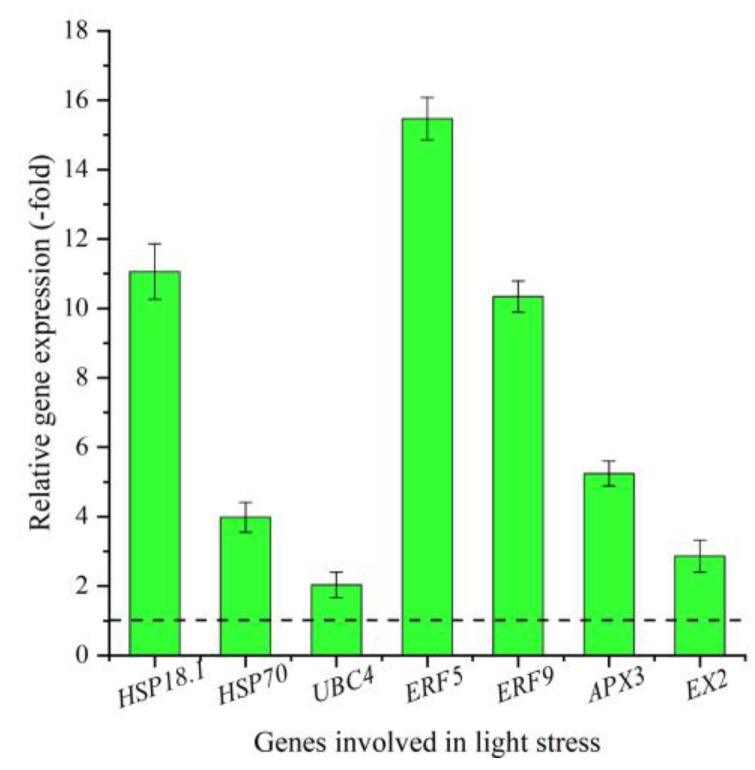
The REL of genes involved in light stress at 3300 m vs. 2300 m, as determined by qRT-PCR (mean ± SD, *n* = 3).

**Figure 5 plants-12-00575-f005:**
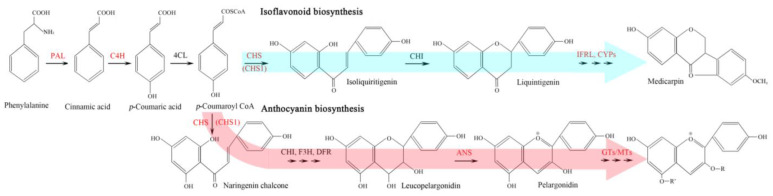
Flavonoid biosynthetic pathway with identified proteins and genes at high altitude shown in red. Enzymatic abbreviates: PAL—phenylalanine ammonia lyase; C4H—cinnamate 4-hydroxylase; 4CL—4-coumarate CoA ligase; CHS—chalcone synthase; CHI—chalcone isomerase; IFRL—isoflavone reductase-like protein; CYP—cytochrome P450; GT—UDP-glycosyltransferase; MT—methyltransferase.

**Figure 6 plants-12-00575-f006:**
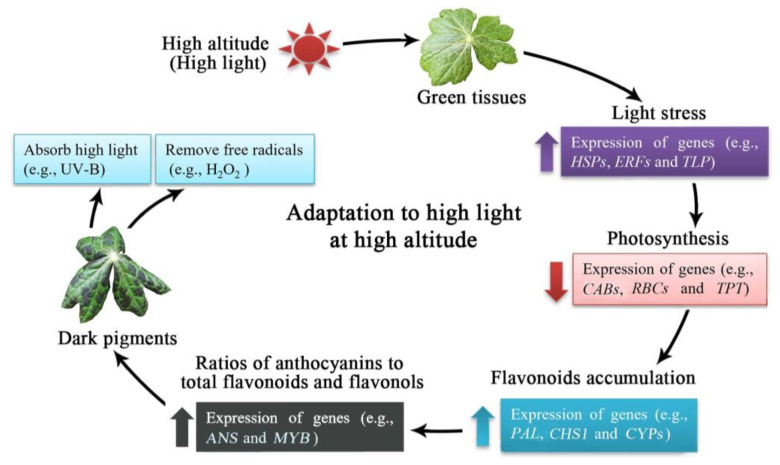
A proposed model of high-altitude-induced flavonoid biosynthesis in *Sinopodophyllum hexdanrum* adaptation to the high altitude that includes high light intensity.

**Figure 7 plants-12-00575-f007:**
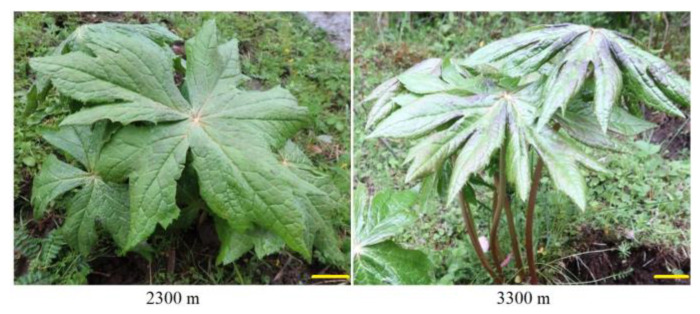
Aerial parts of 3-year-old *S. hexandrum* plants grown at 2300 and 3300 m. Bar represents 2 cm.

**Table 1 plants-12-00575-t001:** Thirty differentially over-expressed proteins at 3300 vs. 2300 m.

Genes	Proteins	No.	Nr ID	Spot Ratio (3300 m vs. 2300 m)
Metabolism (6)				
*BGLU*	Beta-1,3-glucanase	28	gi|458601845	4.57
*ANS*	Anthocyanin synthase	3	gi|575498405	2.51
*CYP6*	Cytochrome P450	25	gi|930809490	2.40
*CYP71BE30*	CYP71BE30	31	gi|441418858	6.09
*PPO1*	Polyphenol oxidase	16	gi|930809512	3.30
*CS*	Corytuberine synthase	14	gi|745698910	2.44
Photosynthesis and energy (10)				
*ycf3*	Photosystem I assembly protein ycf3	9	gi|1021077178	29.30
*rbc*	Ribulose-1,5-bisphosphate carboxylase/oxygenase	2	gi|6513628	2.26
*rbcL*	Ribulose-1,5-bisphosphate carboxylase/oxygenase large subunit	5	gi|825715491	11.33
*petL*	Cytochrome b6-f complex subunit 6	22	gi|1021077124	45.76
*atpA*	ATP synthase CF1 alpha subunit	24	gi|918020571	54.42
*atpB*	ATP synthase CF1 beta subunit	8	gi|918020573	2.39
*atpE*	ATP synthase CF1 epsilon subunit	26	gi|918020601	3.41
*atpA*	ATP synthase subunit alpha	17	gi|728802595	2.92
*matK*	Maturase K	20	gi|356463749	5.73
*infA*	Translation initiation factor IF-1	23	gi|1025806949	2.27
Cell morphogenesis (2)				
*SMCP*	Structural maintenance of chromosomes protein 2	10	gi|913341257	79.04
*ACT*	Actin	15	gi|307147595	9.70
Transcription (4)				
*PI*	PISTILLATA-like protein	1	gi|186909195	2.85
*AP3-1*	APETALA3-like protein 1	7	gi|186909191	5.54
*MYB4*	R2R3-MYB transcription factor MYB4	32	gi|383290957	2.49
*FUL*	FUL-like protein	18	gi|371941952	7.18
Translation (8)				
*rpl16*	Ribosomal protein L16	13	gi|1025806946	4.58
*rpl20*	Ribosomal protein L20	29	gi|1021076983	4.33
*rpl22*	Ribosomal protein L22	19	gi|728802645	6.95
*rps2*	Ribosomal protein S2	21	gi|1025806873	3.47
*rps11*	Ribosomal protein S11	6	gi|1025806951	3.99
*rps12*	Ribosomal protein S12	11	gi|552546114	2.33
*rps14*	Ribosomal protein S14	4	gi|918020607	2.77
*rps16*	Ribosomal protein S16	27	gi|728802592	4.29

Note: the list of proteins is in accordance with the spots in Appendix A.

**Table 2 plants-12-00575-t002:** Twenty-two DEGs involved in flavonoid biosynthesis at 3300 vs. 2300 m.

Genes	Proteins	SwissProt ID	log_2_ FC (3300 vs. 2300 m)
*PAL*	Phenylalanine ammonia-lyase	A0A059XSS3	4.18
*C4H*	Cinnamic acid hydroxylase	Q43054	1.39
*CHS1*	Chalcone synthase 1	P48386	−5.02
*GT6*	UDP-glucose flavonoid 3-O-glucosyltransferase 6	Q9SZG1	−1.94
*IFRL*	Isoflavone reductase-like protein	E1U332	2.07
*BHLH137*	Transcription factor bHLH137	Q93W88	1.72
*DTX41*	Protein DETOXIFICATION 41	Q9LYT3	−1.61
*ABCB19*	ABC transporter B family member 19	Q9LJX0	1.39
*CYP6*	Cytochrome P450	A0A0N9HQE5	3.57
*CYP719A23*	CYP719A23	L7T8H2	1.87
*CYP71CU1*	Cytochrome P450 family 71 subfamily CU polypeptide 1	A0A0N9HTU1	1.39
*CYP734A1*	Cytochrome P450 734A1	O48786	2.64
*CYP80B1*	(S)-N-methylcoclaurine 3′-hydroxylase isozyme 1	O64899	3.49
*CYP81D11*	Cytochrome P450 81D11	Q9LHA1	3.67
*CYP82A2*	Cytochrome P450 82A2	O81972	3.15
*CYP82D47*	Cytochrome P450 CYP82D47	H2DH24	2.23
*CYP94B3*	Cytochrome P450 94B3	Q9SMP5	−1.79
*PPO1*	Polyphenol oxidase	A0A0N9HGV8	4.87
*At5g38780*	S-adenosylmethionine-dependent methyltransferase At5g38780	Q9FKR0	−2.26
*Csa_1G002880*	Protein-L-isoaspartate O-methyltransferase	A0A0A0LRC6	1.51
*HMT1*	Homocysteine S-methyltransferase 1	A4ZGQ8	−2.78
*GXM3*	Glucuronoxylan 4-O-methyltransferase 3	Q9LQ32	1.98

Abbreviation: FC—fold change.

**Table 3 plants-12-00575-t003:** Nineteen DEGs involved in light response at 3300 vs. 2300 m.

Genes	Proteins	SwissProt ID	log_2_FC (3300 vs. 2300 m)
Photosynthesis (9)			
*CAB2*	Chlorophyll a-b binding protein 2	P0CJ48	−2.85
*CAB3*	Chlorophyll a-b binding protein 3	P09756	−2.09
*CAB8*	Chlorophyll a-b binding protein 8	P27490	9.20
*CAB21*	Chlorophyll a-b binding protein 21	P27493	10.72
*RBCS1*	Ribulose bisphosphate carboxylase small chain 1	P16032	−1.73
*RBCS4*	Ribulose bisphosphate carboxylase small chain 4	Q39746	−2.08
*RCAB*	Ribulose bisphosphate carboxylase/oxygenase activase	Q42450	−2.40
*TPT*	Triose phosphate/phosphate translocator	P49132	−2.52
*PSBR*	Photosystem II 10 kDa polypeptide	P06183	1.93
Light stress (10)			
*HSP18.1*	18.1 kDa class I heat shock protein	P27879	7.52
*HSP90-1*	Heat shock protein 90-1	P27323	3.08
*HSP22.0*	22.0 kDa class IV heat shock protein	P30236	2.28
*HSP70*	Heat shock 70 kDa protein	O93866	1.54
*UBC4*	Ubiquitin-conjugating enzyme E2 4	P42748	1.67
*ERF5*	Ethylene-responsive transcription factor 5	O80341	7.18
*ERF9*	Ethylene-responsive transcription factor 9	Q9FE67	4.02
*TLP*	Thaumatin-like protein	Q53MB8	9.28
*APX3*	L-ascorbate peroxidase 3	Q42564	2.22
*EX2*	Protein EXECUTER 2	Q657X6	1.30

Abbreviation: FC—fold change.

## Data Availability

The datasets of transcriptomics are publicly available at NCBI–SRA (http://www.ncbi.nlm.nih.gov/sra (accessed on 1 June 2019)) accession: SRR7196729. The proteomics data have been deposited to the ProteomeXchange Consortium (http://proteomecentral.proteomexchange.org (accessed on 1 March 2017)) with the dataset identifier PXD005640.

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
