# Peer review of "Light-Induced Flavonoid Biosynthesis in Sinopodophyllum hexandrum with High-Altitude Adaptation"

_plants, 2023, doi:10.3390/plants12030575_

Round 1

Reviewer 1 Report

The manuscript entitled “Light-induced flavonoid biosynthesis in Sinopodophyllum hexandrum with high-altitude adaptation” by Zhao et al. aimed to study the effect of high altitude on the flavonoid biosynthesis in Sinopodophyllum hexandrum

The authors observed that the light intensity may correlate with the flavonoid, flavonol, and anthocyanin content. To quantitatively investigate the mechanisms, the authors combine HPLC, proteomics  with transcriptomics to study the amount and type of flavonoids, differentially expressed proteins (DEP)/genes (DEG) at 2300 m and 3300 m, respectively. And they found 9 DEPs and 41 DEGs, which may indicate the regulatory role of light intensity to adapt to high altitude and high light-intensity. 

Overall, the conclusions are appropriate, and supported by the data. I recommend accepting it. 

Author Response

Thanks for your reviewing and approving our researches.

Reviewer 2 Report

The manuscript by Zhao et al describes the effect of high-altitude adaptation on flavonoid biosynthesis in Sinopodophyllum hexandrum. For that, authors analysed the flavonoid content, as well as the abundance of proteins and expression of genes related to flavonoid metabolism in Sinopodophyllum hexandrum plants grown at high altitude (2300 and 3300 metres). The manuscript is well written, and it includes interesting results from the perspective of both basic plant research and possible agricultural application. However, before it can be accepted for application few corrections should be implemented.

My main concern is that factors other than light intensity are mostly ignored when analysing the results from this study. Precipitation and, specially, temperature are going to vary wildly in the 1000 metres of difference between the two sites. Firstly, climatic data from the two sites used in this study must be provided, in addition to a description of the soil composition and characteristics. Secondly, this climatic data must be integrated in the discussion, as changes in light intensity are not going to be the only factors governing the changes observed by the authors. The effect of temperature in particular, although briefly mentioned by the authors, should be expanded in the discussion. Among other effects, the lower temperatures expected in the higher altitude site are probably going to negatively affect the dark phase of the photosynthesis, therefore exacerbating the photo-damage and ROS production caused by hight light intensity.

In addition, the experimental setup and plant cultivation should be better described. Even though the authors provide a citation of a previous work, I consider that this kind of basic information should be explicitly included in the manuscript. At least the methods for seed germination and seedling establishment and transplantation (if any) should be described. In the citation provided by the authors plants were sown at one altitude (either 2300 or 3300 m) and then transplanted to the other; is that also the case here? Also, if plants were sown in May 2013, when the samples were collected in July 2016 said plants would have been 3 years (and 2 months) old, not 4 years old.

These are some other points that should be corrected:

Line 110: “Metabolism” is a way too general term, particularly compared with the rest of the list. Authors should be more specific. Additionally, the summatory of proteins in this list only goes to 30, while authors mention 64 differentially expressed proteins. What is the function of the other 34? Or were those proteins with unknown function?

L117. I suggest rephrasing. It does not look like the use of “Based on” is convenient here.

L134. The table 2 does not seem to include Reads Per kb per Million values, but rather ratios between expression and both altitudes. I would suggest rephrasing. The same goes for L145.

L142. Unless they are included in a table (not in supplementary materials), the full name of the proteins and genes should be included the first time they are mentioned.

L181-202. This whole paragraph is just describing know biosynthetic pathways, without discussing their own data. It should be heavily reduced and moved to the introduction, or completely deleted.

L203. These proteins and genes may be affected by altitude in a similar way as flavonoid biosynthesis, but that does not necessarily imply that they are directly involved in flavonoid biosynthesis. Unless authors can provide proof of their involvement in flavonoid biosynthesis, this sentence should be rephrased.

L247. As in the previous comment, I do not think that the data shown in this manuscript proves that the upregulation of these genes is responsible for increased flavonoid biosynthesis.

Figure 6. I am not sure if I follow the logic behind the circular model, since the increased capacity to absorb UV light and remove free radicals would not “feedback” into the high altitude (or high light), which is just an environmental condition. Maybe authors should consider the use of a linear model. Also, I am not sure that is correct to imply that increase in anthocyanin/flavonoid ratio is a consequence of the increase in flavonoid accumulation.

L406. Please, specify the statistical test used.

L411. Not all of these parameters were enhanced by the up-regulation of genes and proteins in flavonoid biosynthesis. The higher ratios of anthocyanins to total flavonoids and flavonols was provably not enhanced by increased flavonoid biosynthesis (unless authors can prove that increased flavonoid biosynthesis leads directly to an even greater increase in anthocyanin biosynthesis), and the higher light intensity definitely did not (as it is an environmental variable).

Author Response

Thanks very much for all your comments that are helpful to improve our paper much better. We have tried to address and correct each comment. Attachments below with our responses are shown in bold. Revised parts are marked up using the “Track Changes” function in the manuscript. 

1. My main concern is that factors other than light intensity are mostly ignored when analysing the results from this study. Precipitation and, specially, temperature are going to vary wildly in the 1000 metres of difference between the two sites. Firstly, climatic data from the two sites used in this study must be provided, in addition to a description of the soil composition and characteristics. Secondly, this climatic data must be integrated in the discussion, as changes in light intensity are not going to be the only factors governing the changes observed by the authors. The effect of temperature in particular, although briefly mentioned by the authors, should be expanded in the discussion. Among other effects, the lower temperatures expected in the higher altitude site are probably going to negatively affect the dark phase of the photosynthesis, therefore exacerbating the photo-damage and ROS production caused by high light intensity.

Thanks for your suggestion, firstly, the environmental parameters and soil characteristics were shown in Table S1 and Table S2, respectively. (Page 9, lines 279-280)

Secondly, the effect of temperature on plant growth and metabolites accumulation has been expanded in the Discussion section: “Specifically, low temperatures (4-15°C) could enchane plant biomass and accumulation of secondary metabolites, especially in podophyllotoxin (PPT), in S. hexandrum plants by the upregulation of genes involved in plant growth, PPT biosynthesis, and stress response [16-18]; a moderate water deficit could improve PPT accumulation [19]; a moderately shaded habitat was in favor of plant growth [20]; and UV-B radiation could inhibit plant growth and PPT biosynthesis while improve flavonoids accumulation [21].” (Page 7, lines 172-178)

Additionally, your excellent state: “Among other effects, the lower temperatures expected in the higher altitude site are probably going to negatively affect the dark phase of the photosynthesis, therefore exacerbating the photo-damage and ROS production caused by high light intensity.” has been added in the Discussion section. (Page 7, lines 178-181)

2. In addition, the experimental setup and plant cultivation should be better described. Even though the authors provide a citation of a previous work, I consider that this kind of basic information should be explicitly included in the manuscript. At least the methods for seed germination and seedling establishment and transplantation (if any) should be described. In the citation provided by the authors plants were sown at one altitude (either 2300 or 3300 m) and then transplanted to the other; is that also the case here? Also, if plants were sown in May 2013, when the samples were collected in July 2016 said plants would have been 3 years (and 2 months) old, not 4 years old.

Thanks for your suggestion, the basic information about the seed germination, seedling establishment and transplantation has been provided: “The seeds separated from the fruit were pre-treated with 0.01% Bavistin for 10 min, immersed in sterile 500 mg/L GA3 for 48 h, and then sown at the 2300 and 3300 m sites in May, 2013. Some seedlings were transplanted to Weiyuan (2300 m; 34°58′8”N, 104°04′52”E) of Gansu province in May, 2014 [3].” (Page 9, lines 275-279)

Additionally, the plant age of S. hexandrum plants has been revised to “3-year-old plants”. (Pages 9 and 10, lines 281 and 291).

These are some other points that should be corrected:

3. Line 110: “Metabolism” is a way too general term, particularly compared with the rest of the list. Authors should be more specific. Additionally, the summary of proteins in this list only goes to 30, while authors mention 64 differentially expressed proteins. What is the function of the other 34? Or were those proteins with unknown function?

According to your comments, the description of “Metabolism” has been revised to: “primary metabolism and secondary metabolism”. (Page 4, lines 116)

Additionally, the description about the summary of proteins has been revised to: “A total of 65 proteins were differentially expressed at 3300- vs. 2300-m (Figure S1) with 46 over-expressed and 19 down-expressed (Figure S1); among the 46 over-expressed proteins, 30 proteins had known function while the other 16 proteins had no known function (Table 1).”. (Page 3, lines 112-115)

4. L117. I suggest rephrasing. It does not look like the use of “Based on” is convenient here.

According to your comments, the sentence has been rephrased to: “Transcriptomic data of plants at 3300- vs. 2300-m [36] showed that 22 genes encoding for flavonoid catalysis were differentially expressed with 16 genes up-regulated 1.4 (C4H, ABCB19 and CYP71CU1) to 4.9-fold (PPO1), while other genes (6) were down-regulated by 1.6 (DTX41) to 5.0-fold (CHS1) (Table 2)”. (Page 4, lines 124-128)

5. L134. The table 2 does not seem to include Reads Per kb per Million values, but rather ratios between expression and both altitudes. I would suggest rephrasing. The same goes for L145.

According to your comments, the description about the “Reads Per kb per Million values” has been revised to: “the fold change (FC) at the two altitudes”. (Pages 5 and 6, lines 142 and 154)

6. L142. Unless they are included in a table (not in supplementary materials), the full name of the proteins and genes should be included the first time they are mentioned.

According to your comments, the Table S1 has been moved to the text as Table 3, which shows not only the full name but also the specific information of the proteins and genes. (Page 6, line 158)

7. L181-202. This whole paragraph is just describing know biosynthetic pathways, without discussing their own data. It should be heavily reduced and moved to the introduction, or completely deleted.

Thanks for your suggestion, the whole paragraph about flavonoids biosynthesis has been completely deleted, since the biosynthetic pathways of flavonoids has been known, meanwhile this can reduce the repetition rate.

8. L203. These proteins and genes may be affected by altitude in a similar way as flavonoid biosynthesis, but that does not necessarily imply that they are directly involved in flavonoid biosynthesis. Unless authors can provide proof of their involvement in flavonoid biosynthesis, this sentence should be rephrased.

According to your comments, the sentence: “In this study, 5 proteins (ANS, CYP6, CYP71BE30, PPO1 and MYB4) and 22 genes (PAL, C4H, CHS1, IFRL, GT6, BHLH137, DTX41, ABCB19, 9 CYPs, PPO1 and 4 MTs) were identified to be involved in flavonoid biosynthesis.” has been rephrased to: “In this study, 5 proteins (ANS, CYP6, CYP71BE30, PPO1 and MYB4) and 22 genes (PAL, C4H, CHS1, IFRL, GT6, BHLH137, DTX41, ABCB19, 9 CYPs, PPO1 and 4 MTs) may be involved in flavonoid biosynthesis.” (Page 7, lines 202-204)

9. L247. As in the previous comment, I do not think that the data shown in this manuscript proves that the upregulation of these genes is responsible for increased flavonoid biosynthesis.

According to your comments, the sentence has been rephrased to: “In this study, the up-regulation of selected genes (HSP18.1, HSP70, UBC4, ERF5, ERF9, APX3 and EX2) may play important roles in enhancing flavonoid biosynthesis and protecting plants from abiotic stress (e.g., high light) at high altitudes.” (Page 9, lines 250-252)

10. Figure 6. I am not sure if I follow the logic behind the circular model, since the increased capacity to absorb UV light and remove free radicals would not “feedback” into the high altitude (or high light), which is just an environmental condition. Maybe authors should consider the use of a linear model. Also, I am not sure that is correct to imply that increase in anthocyanin/flavonoid ratio is a consequence of the increase in flavonoid accumulation.

Thanks for your comments, the arrows in the final step involved in the “feedback” into the high altitude (or high light) has been deleted from the Figure 6. (Page 9, line 268)

Indeed, you are right to doubt about that the increase in anthocyanin/flavonoid ratio is a consequence of the increase in flavonoid accumulation. As known, the total flavonoids include: flavonols, flavones, isoflavones, anthocyanidins, flavanones, flavanols, and chalcones. In this study, the contents of total flavonoids, three flavonols (i.e., isoquercetin, quercetin and kaempferol), and anthocyanins were determined, as well as the expression level of genes involved in biosynthesis of flavonoids and anthocyanins were quantified. Thus, the proposed step was implied according to our findings.

11. L406. Please, specify the statistical test used.

According to your comments, the statistical test used in this study has been specified: “All measurements were performed using three biological replicates. Statistical analysis was conducted via a t-test for independent samples. SPSS 22.0 was the software package used with P<0.05 as the basis for statistical differences.” (Page 12, lines 414-417)

12. L411. Not all of these parameters were enhanced by the up-regulation of genes and proteins in flavonoid biosynthesis. The higher ratios of anthocyanins to total flavonoids and flavonols was probably not enhanced by increased flavonoid biosynthesis (unless authors can prove that increased flavonoid biosynthesis leads directly to an even greater increase in anthocyanin biosynthesis), and the higher light intensity definitely did not (as it is an environmental variable).

According to your comments, the description about the parameters in Conclusion section has been revised to: “From the above observations, greater contents of total flavonoids, flavonols and anthocyanins in S. hexandrum were observed at higher altitudes, which may be regulated by the up-regulated genes or over-expressed proteins involved in flavonoid biosynthesis and light stress.” (Page 13, lines 419-423)

Moreover, the English language and style have been carefully checked throughout the text.